# Enforcing 3D Coherence in Semi-Supervised Segmentation for Pancreatic Tumor Histopathology from Light Sheet Fluorescence Microscopy

**Yousif Hashisho**[1]    YOUSIF.HASHISHO@PLANET-AI.DE
**Diana Pinkert-Leetsch**[2]    DIANA.PINKERT-LEETSCH@MED.UNI-GOETTINGEN.DE
**Jeannine Missbach-Guentner**[2]    J.MISSBACH@MED.UNI-GOETTINGEN.DE

[1] *PLANET AI, Rostock, Germany*

[2] *University Medical Center Göttingen, Department of Clinical and Interventional Radiology & Department of Cardiovascular Imaging, Göttingen, Germany*

**Editors:** Accepted for publication at MIDL 2026

## Abstract

Light sheet fluorescence microscopy (LSFM) provides unprecedented two-dimensional (2D) tomographic views and three-dimensional (3D) reconstructions of tissue volumes, but generates such large data sets that complete annotation is not feasible. This results in volumes with sparse axial annotations, where ground truth is available for only a small fraction of slices. Standard semi-supervised learning (SSL) methods often fail in this regime, unable to bridge the large gaps between labeled slices to produce coherent 3D segmentations. To address this, we propose a novel SSL framework designed to enforce 3D anatomical plausibility from sparse 2D supervision. The core of our contribution is an axial continuity loss, a regularization term that enforces prediction consistency between adjacent unlabeled slices. This loss is integrated into a voxel-aware Mean-Teacher framework that effectively leverages abundant unlabeled data. We validate our approach on a 3D LSFM dataset of human pancreatic ductal adenocarcinoma (PDAC), which we collected and sparsely annotated for this study. Our experiments show that standard SSL baselines degrade in performance as annotations become sparser, producing noisy predictions between labeled slices. In contrast, our full framework, which integrates an attention-gated 3D U-Net with our proposed continuity loss, maintains robust 3D coherence even in low-data regimes, enabling reliable histopathological analysis from minimal annotations.

**Keywords:** Semi-Supervised Learning, Light Sheet Fluorescence Microscopy, Sparse Annotation, 2D and 3D Segmentation, Pancreatic Cancer, Continuity Loss.

## 1. Introduction

Pancreatic ductal adenocarcinoma (PDAC) is a highly lethal malignancy with a five-year survival rate below 10%, and the numbers are steadily rising (Lippi and Mattiuzzi, 2020). Current approaches combine multimodal imaging with AI-based deep learning to enhance histopathological evaluation (Levy et al., 2024; van Diest et al., 2024), yet these methods rely on large, high-quality, and expertly annotated datasets. Light sheet fluorescence microscopy (LSFM) provides 3D tomographic data of entire tissue volumes, enabling precise structural analysis beyond the limitations of conventional planar sections. Here, 3D microscopy with its tomographic datasets offers an ideal basis and additional advantages for comprehensive data analysis (Song et al., 2024; Liu et al., 2024). The development of a reliable segmentation

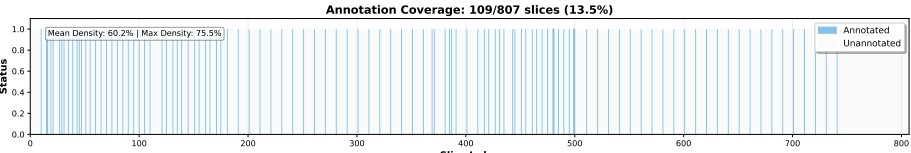

(a) Annotation coverage across the 3D volume.

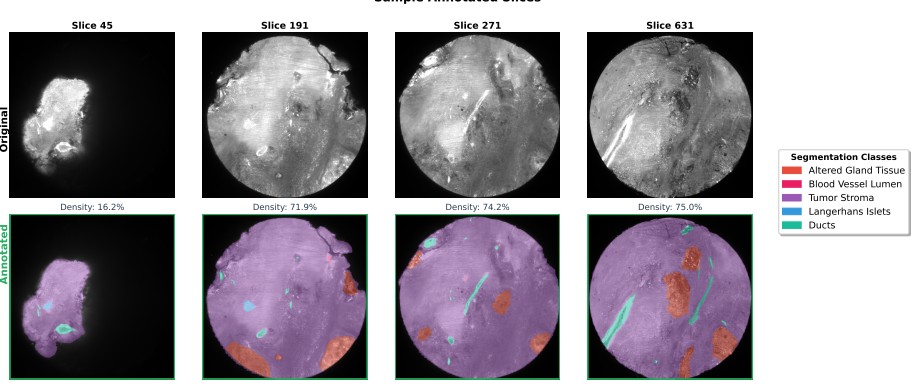

(b) Sample annotated slices with varying annotation density.

Figure 1: **Illustration of the Sparse Annotation Challenge.** (a) Annotation coverage across the 807 axial slices of a representative PDAC volume, showing that only 109 slices (13.5%) contain any ground-truth labels. (b) Detailed view of four representative annotated slices, showing the raw LSFM image (top row) and the same slice with ground-truth segmentation masks overlaid (bottom row).

method enabled rapid analysis of tumor tissue with regard to the heterogeneity of PDAC tumors, which is prospectively crucial for the development of effective therapies and the improvement of patient survival rates.

While LSFM offers a holistic view, its potential is currently bottlenecked by the lack of automated analysis tools. Although deep learning has transformed digital pathology (Levy et al., 2024; van Diest et al., 2024; Campanella et al., 2019; Bulten et al., 2022), traditional 2D histology relies on manual examination of thin slices, a process that inherently discards 3D spatial context and is prone to sampling bias. To fully exploit LSFM data, we need dense 3D segmentation of tissue structures. However, manually annotating these terabyte-scale volumes is prohibitively time consuming. Consequently, datasets are often sparsely annotated, with ground truth available only on a few scattered 2D slices, leaving the vast majority of the 3D volume unlabeled. This sparsity poses a major challenge: standard semi-supervised learning (SSL) methods struggle to generalize across the large, unstructured gaps between labeled slices, often producing incoherent and anatomically implausible 3D predictions.

To address this challenge, we propose a novel SSL framework designed to learn coherent 3D structures from sparsely annotated volumes. Our key contributions are:

- We introduce and validate our method on a newly created 3D LSFM dataset of human PDAC volumes, which were collected and sparsely annotated by histological experts.

- We propose an axial continuity loss, a regularization term that enforces prediction consistency between adjacent unlabeled slices, providing a crucial inductive bias for 3D coherence.

- We integrate this loss into a voxel-aware Mean-Teacher framework and employ a 3D U-Net with an attention-gated decoder to capture complex, long-range spatial dependencies.

We demonstrate the efficacy of our approach by showing that while a strong Mean-Teacher SSL baseline degrades as annotations become sparser, the addition of our proposed axial continuity loss provides the necessary 3D inductive bias to maintain robust performance and anatomical coherence.

## 2. Related Work

The high cost of expert annotation has made SSL a cornerstone of medical image analysis. Our framework builds upon the Mean-Teacher model (Tarvainen and Valpola, 2017), which enforces consistency between a student and a teacher network. Recent advancements have refined this by decoupling training to filter unreliable pseudo-labels (Das et al., 2024) or leveraging multi-modal consistency (Zhou et al., 2024).

A specific challenge in 3D imaging is learning dense volumes from sparse slice annotations. Approaches like cross-teaching (Cai et al., 2023) leverage 2D networks to guide 3D models, while others bootstrap dense predictions from initial 2D outputs (Thiyagarajan et al., 2024). To enforce anatomical plausibility, prior works have employed inter-slice context modules (Zhang et al., 2020) or smoothness losses (Wu et al., 2022). Our work distinguishes itself by introducing an axial continuity loss directly within the SSL objective. Unlike complex architectural additions, our loss acts as a simple, voxel-wise geometric prior that synergizes with attention mechanisms to enforce local 3D coherence in the absence of dense supervision.

## 3. Materials and Methods

### 3.1. Dataset Curation and Imaging

Human specimens of pancreatic lesions were obtained from the Department of Pathology, University Medical Center (Göttingen, Germany). The study was performed according to the guidelines of the local ethics committee of the University Medical Center Göttingen (permission number 05/10/17) and in accordance with the declaration of Helsinki.

Punch biopsies with a diameter of 3 mm were taken from the paraffin-embedded pancreatic tissue and prepared for LSFM as described before (Sagar et al., 2024).

LSFM was performed using the UltraMicroscope Blaze™ (Miltenyi Biotec B.V.& Co KG, Germany). The specimens were attached either by a screw or with superglue to the sample holder of the microscope. Fluorescence imaging was done using an NKT SuperK Extreme white light laser of 0.6 W visible power (NKT Photonics A/S, Denmark) as a

light source. The corresponding filter sets for excitation *(ex)* and emission *(em)* used are 470 $nm_{ex}$/525 $nm_{em}$; 560 $nm_{ex}$/620 $nm_{em}$ and 710 $nm_{ex}$/810 $nm_{em}$. The standard measurement parameters include a numerical aperture of 0.163 and a light sheet thickness of 3.9 $\mu$m. Using the 4x objective, a sample step size in Z-direction of 4 $\mu$m the pixel size was 1.6 $\mu$m. Images were captured using the ImSpector (version 7.5.2) software. The Blaze™ is equipped with a 4.2-megapixel sCMOS camera with a 2048 × 2048 pixel resolution (PCO AG, Germany). LSFM data analysis was performed using Zeiss arivis Pro software (Carl Zeiss Microscopy Software, Germany).

For histological validation, the biopsies were embedded in paraffin again and sliced into 2 $\mu$m sections for further staining, as described earlier (Pinkert-Leetsch et al., 2025). To streamline the annotation process, we utilized an AI-assisted workflow described in Appendix A.1. Further details on dataset characteristics and class distribution are provided in Appendix A.2.

## 4. Semi-Supervised Segmentation Framework

To address the challenge of sparse axial annotations, we developed a semi-supervised learning framework designed to enforce 3D coherence. Our approach integrates a consistency-based Mean-Teacher paradigm with an axial continuity loss, trained within an end-to-end deep segmentation model.

### 4.1. Mean-Teacher Framework

We adopt the Mean-Teacher approach, a powerful consistency-based SSL paradigm (Tarvainen and Valpola, 2017). The framework consists of a student model and a teacher model of identical architecture. The student's weights ($\theta_s$) are updated via standard backpropagation, while the teacher's weights ($\theta_t$) are an Exponential Moving Average (EMA) of the student's weights: $\theta_t \leftarrow \alpha\theta_t + (1 - \alpha)\theta_s$. The high EMA decay rate $\alpha$ ensures the teacher's updates are smooth and stable.

During training, the student receives a strongly augmented version of the input data, while the teacher receives a weakly augmented version. The teacher then generates pseudo-labels for the unlabeled regions, which the student is trained to match. This consistency enforcement, a widely used technique in SSL (Sohn et al., 2020), provides a powerful learning signal from the abundant unlabeled data.

### 4.2. Composite Loss Function

The total loss, $L_{total}$, is a weighted sum of a supervised loss on labeled data ($L_{sup}$), an unsupervised consistency loss on unlabeled data ($L_{cons}$), and our axial continuity loss ($L_{cont}$).

$$L_{total} = L_{sup} + \lambda_{cons}L_{cons} + \lambda_{cont}L_{cont} \tag{1}$$

#### 4.2.1. SUPERVISED AND CONSISTENCY LOSSES

For labeled pixels, we compute a standard supervised segmentation loss using a combination of Dice loss (Milletari et al., 2016) and Cross-Entropy (DiceCE). For unlabeled pixels, we enforce consistency between the student's predictions and the teacher's pseudo-labels.

The teacher's predictions are sharpened using temperature scaling ($T < 1.0$), a technique widely used in self- and semi-supervised learning to control the sharpness of probability distributions (Chen et al., 2020; Xie et al., 2020). Crucially, our framework applies voxel-aware consistency: rather than enforcing image-level agreement, we filter teacher pseudo-labels at the voxel level using a confidence threshold $\tau$, so that only high-confidence voxel predictions contribute to the consistency loss. The consistency loss, $L_{cons}$, is then calculated as the Kullback-Leibler (KL) divergence between the student's output and the filtered, sharpened teacher pseudo-labels.

### 4.2.2. Axial Continuity Loss ($L_{cont}$)

The core of our contribution is the axial continuity loss, designed to enforce 3D coherence across unlabeled regions. While spatial consistency has been used as a regularizer in various contexts, such as in Conditional Random Fields (Krähenbühl and Koltun, 2011) or for temporal smoothing in video analysis, our formulation specifically targets the challenge of propagating information across sparse axial annotations in an SSL framework.

This loss acts as an inductive bias, encouraging the model to produce anatomically plausible segmentations by ensuring predictions on adjacent slices are similar. For a given unlabeled slice $z_i$, the loss penalizes the difference between its prediction, $P_i$, and the prediction of its adjacent unlabeled slice, $P_{i+1}$:

$$L_{cont} = \frac{1}{N_{unlabeled}} \sum_{i \in \text{unlabeled}} \mathcal{D}(P_i, P_{i+1}) \tag{2}$$

where $\mathcal{D}$ is the Mean Squared Error (MSE) between the softmax probability distributions of the adjacent slice predictions. This regularization is applied only between pairs of adjacent slices that are both unlabeled, as shown in Figure 2, preventing interference with the supervised signal at the boundaries of annotated regions. This simple yet effective mechanism allows the model to propagate learned structural information from the sparse labeled slices throughout the entire volume.

### 4.3. Model Architecture

Our segmentation model is based on the 3D U-Net architecture (Çiçek et al., 2016), a standard encoder-decoder with skip connections. While effective for local feature extraction, the sparse nature of our annotations requires the model to learn relationships between distant features, effectively propagating information from a few labeled slices across large unlabeled volumes.

To address this, we enhance the standard U-Net decoder to better capture these long-range spatial dependencies. We integrate Sequential Axial Attention blocks (Wang et al., 2020) into the two deepest stages of the decoder. These blocks apply self-attention sequentially along the depth, height, and width axes, allowing the model to aggregate features from a global receptive field. By placing these blocks in the decoder's bottleneck, where the feature maps are smallest, we achieve maximum contextual reach with minimal computational overhead. This architectural choice explicitly encourages the model to leverage global context from the entire input volume when making predictions, which is critical for generating coherent segmentations from sparse supervisory signals.

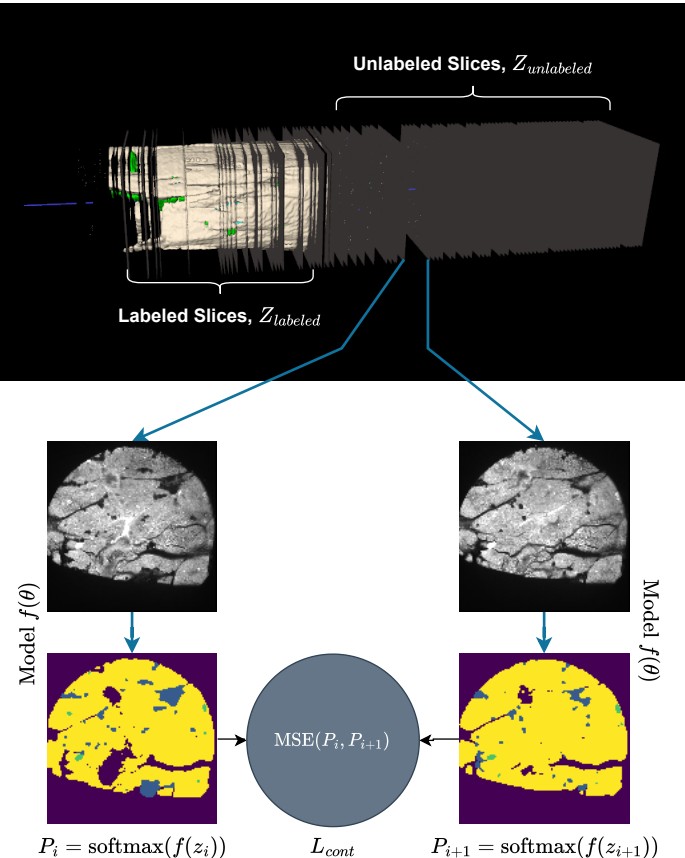

Figure 2: Illustration of the Axial Continuity Loss. The loss penalizes inconsistencies between the predicted probability maps $(P_i, P_{i+1})$ of adjacent unlabeled slices $(z_i, z_{i+1})$, encouraging smooth and coherent 3D segmentations.

The decoder then symmetrically upsamples the feature maps, concatenates them with skip connections from the encoder, and passes them through these attention-gated stages before a final 1x1 convolution produces the voxel-wise classification. The impact of these attention blocks is quantified in our ablation studies in Section 5.3.

### 4.4. Training and Evaluation Strategy

As established in Appendix A.2 and Appendix A.3, our dataset presents a key challenge: not all histopathological classes are present in every volume. A traditional train/validation split (e.g., using two volumes for training and one for validation) is therefore infeasible, as it would result in a validation set containing classes never seen during training.

To address this, we adopt a unified training approach where all three volumes are included in the training set, ensuring the model is exposed to all available classes. Our evaluation strategy is then based on the model's ability to generalize to unseen data within

these volumes. We achieve this by deterministically holding out a fraction of the annotated slices from the training process. These held-out annotated slices serve as our validation set, allowing us to quantify the model's performance on its primary task: accurately segmenting sparsely distributed, unseen ground truth, by leveraging the surrounding unlabeled 3D context.

## 5. Experiments and Results

We conducted a series of experiments to validate our proposed framework, dissecting the individual and combined contributions of our axial continuity loss and the attention-gated decoder.

### 5.1. Experimental Setup

**Models.** We compare four configurations on our primary anisotropic dataset: (1) an SSL Baseline with a standard 3D U-Net; (2) the baseline augmented with only attention blocks; (3) the baseline with only our continuity loss; and (4) our full Proposed Method combining both. All methods share the same hyperparameters.

**Implementation Details.** Our framework is implemented in PyTorch. We train for 1000 epochs using the AdamW optimizer with a cosine learning rate scheduler (initial LR 1e-4) and an EMA decay $\alpha = 0.999$. The consistency and continuity loss weights ($\lambda_{cons}$, $\lambda_{cont}$) are ramped up to 10.0. We use a temperature $T = 0.5$, confidence threshold $\tau = 0.8$, and train on random anisotropic patches (64x128x128) with foreground oversampling.

**Evaluation Metrics.** All models are evaluated on the held-out set of annotated validation slices. We report the per-class and mean Dice Similarity Coefficient (DSC).

### 5.2. Synergistic Effect of Attention and Continuity Loss

Our primary evaluation, summarized in Table 1, reveals a crucial synergistic relationship between our two main contributions. We first established a strong, attention-enabled baseline (Baseline + Attention) which achieves a mean DSC of 0.86. Our main comparison is against this powerful baseline.

As shown in the table, our full Proposed Method, which integrates the continuity loss, further improves performance to a final mean DSC of 0.88. While the overall improvement is modest, a per-class analysis reveals the critical nature of this synergy. The combination of global context from the attention blocks and local coherence from the continuity loss unlocks substantial performance gains on several of the most challenging and rarest classes. We observe a 5-point increase for *Altered Gland Tissue* and a 3-point increase for *Langerhans Islets*. This demonstrates that our full framework provides a powerful regularization that specifically benefits the segmentation of small, complex structures that are difficult to learn from sparse supervision alone.

### 5.3. Ablation Studies

**Component Contribution Analysis.** To fully understand the interplay between our contributions, Table 2 details the performance of all four model configurations. The re-

Table 1: Per-class Dice Similarity Coefficient (DSC) on the held-out validation slices. The primary comparison is between the strong attention-based baseline and our full proposed method. The synergy of continuity and attention yields significant gains on challenging, rare classes.

| Histopathological Class | Baseline + Attention (DSC) | Proposed (Cont. + Attn.) (DSC) |
|---|---|---|
| Altered Gland Tissue | 0.82 | **0.87** |
| Fat Tissue | **0.88** | 0.87 |
| Blood Vessel Lumen | 0.80 | **0.82** |
| Tumor Stroma | **0.98** | **0.98** |
| Langerhans Islets | 0.81 | **0.84** |
| Ducts | 0.79 | **0.81** |
| Gland Tissue | 0.96 | **0.97** |
| **Mean (All Classes)** | **0.86** | **0.88** |

sults reveal a critical synergistic relationship. Adding attention alone provides a significant performance boost (+1 point over the standard SSL Baseline). In contrast, adding the continuity loss in isolation to the standard U-Net results in a slight performance degradation (-1 point). The best results are only achieved when both components are used together, suggesting that the robust, global features generated by the attention blocks are a prerequisite for the continuity loss to effectively regularize the model.

Table 2: Full component ablation study on the anisotropic dataset (mean DSC), demonstrating the synergistic effect of combining attention and continuity loss.

| Method Configuration | Mean DSC |
|---|---|
| SSL Baseline (Standard U-Net) | 0.85 |
| + Axial Continuity Loss Only | 0.84 |
| + Attention Blocks Only | 0.86 |
| **Proposed Method (Continuity + Attention)** | **0.88** |

### 5.4. Qualitative Results

To provide qualitative insight into model performance, we visualize the segmentation outputs for our two key scenarios: the extreme sparsity case and our primary full-depth dataset (Figure 3).

**Performance in Extreme Sparsity.** The top row of the figure demonstrates the baseline's failure to generalize in the extreme sparsity scenario. Given an unlabeled input slice from deep within the volume, the SSL Baseline's prediction is incoherent and appears random. This indicates that without sufficient axial context or explicit regularization, the

| Input Slice | Baseline | Continuity |
|:-:|:-:|:-:|

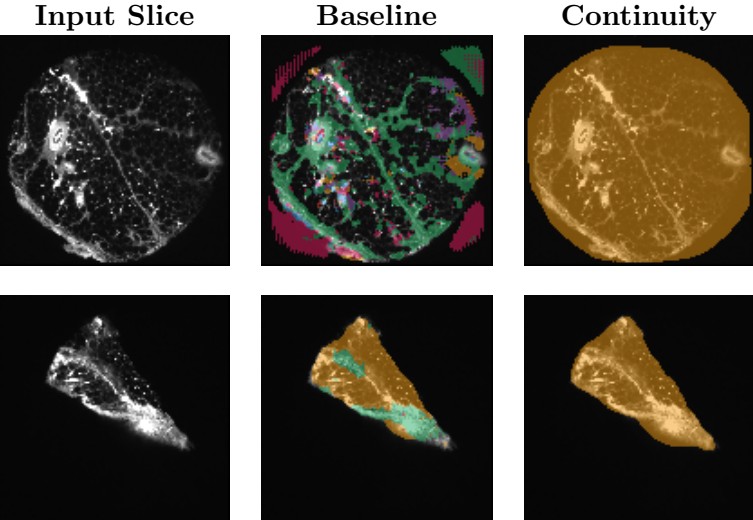

Figure 3: **Qualitative Comparison on Unlabeled Slices.** The top row shows results from models trained on the severely downsampled (extreme sparsity) dataset, while the bottom row shows results from models trained on our primary full-depth dataset. In both scenarios, the addition of the continuity loss provides a clear benefit.

model fails to learn meaningful anatomical features from the consistency objective alone. In stark contrast, our method with the continuity loss produces a coherent and anatomically plausible segmentation from the same input, successfully interpolating structural information across the large unlabeled gaps.

The bottom row compares the models on our primary full-depth dataset, focusing on a challenging slice near the edge of the volume where boundary definition is difficult. Here, the baseline model (without continuity loss) is confused by high-intensity artifacts, producing several false positive segmentations within the *Fat Tissue*. Our full method, which incorporates the continuity loss, is robust to these artifacts. It correctly identifies the entire region as a single tissue type, producing a cleaner and more accurate segmentation that demonstrates superior boundary definition.

For a more extensive, slice-by-slice comparison of ground truth and model predictions for each volume, please see Appendix C.

## 6. Discussion and Conclusion

This study addresses the critical bottleneck of annotation sparsity in 3D LSFM histopathology. Our results demonstrate that while standard consistency-based SSL methods fail to bridge large axial gaps, introducing an explicit axial continuity loss successfully enforces 3D anatomical plausibility.

Technically, the continuity loss functions as a geometric prior, penalizing high-frequency z-axis variations characteristic of model collapse in sparse regimes. This is evident in our extreme sparsity experiments, where the baseline produced incoherent noise while our method maintained structural integrity. A hallucination analysis (Appendix B) further confirms this: the continuity constraint reduces the overall hallucination rate, slices where a class is predicted but absent in the ground truth, from 8.88% to 0.98%, demonstrating that it suppresses spurious, inconsistent predictions rather than propagating them. Crucially, our ablation study reveals a synergy between this local regularization and the global context provided by the attention-gated decoder. We hypothesize that attention enables the model to build robust semantic representations, which the continuity loss then refines into spatially coherent shapes. This combination proved especially beneficial for rare classes like *Altered Gland Tissue* and *Langerhans Islets*.

Clinically, this framework paves the way for scalable quantitative analysis of large-scale LSFM datasets. By reducing the annotation burden to a small fraction of slices, we enable the routine extraction of 3D morphometric features—such as tumor tissue volume and vascular connectivity—that are inaccessible to traditional 2D histology.

Our study is limited by the use of three patient samples, a constraint often inherent to novel imaging modalities. Future work will focus on validating this approach on larger, multi-center cohorts and investigating its applicability to other volumetric imaging tasks where annotation is expensive. The tomographic LSFM based images of human pancreatic malignancies provided thousands of sequential image layers, which are ideal for training AI-based tools and thus contribute significantly to automation, saving time, resources, costs, and personnel in diagnostic reporting in the future. Furthermore, the approach presented here will lead to added diagnostic value compared to planar histology alone, even without additional staining, enabling the identification of novel imaging tumor markers, and thus a more refined analysis of the organ and tumor interstitium.

## Acknowledgments

We thank Sabine Wolfgramm, Bettina Jeep and Julia Fascher for excellent technical assistance in clearing and embedding procedures, and for performing histology of tissue sections. Furthermore we thank Helge Lange, Gundram Leifert, Archit Anwai and Constantin Pape for their collegial exchange and valuable support. The project was supported by the Bundesministerium fuer Forschung, Technologie und Raumfahrt, Deutschland (Federal Ministry of Research, Technology and Space, Germany) CAPTAIN: 13GW0647D to J.M.G., University Medical Center Goettingen, Germany and 13GW0647B to Y.H. H.L. and G.L., Planet AI, Rostock, Germany.

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

## Appendix A. Extended Dataset Details

### A.1. AI-Assisted Annotation Workflow

To manage the large-scale annotation required for this study, we established an efficient, semi-automated "human-in-the-loop" workflow using the open-source Computer Vision Annotation Tool (CVAT) platform.

To accelerate the time-intensive task of manual segmentation, we integrated interactive segmentation models into the CVAT environment. Foundational models like the Segment Anything Model (SAM) (Kirillov et al., 2023) have revolutionized this space by enabling precise mask generation from simple user prompts. We leveraged the latest advancements in this line of work by deploying Meta AI's SAM 2 alongside a domain-adapted Segment Anything for Microscopy (SAM-M) model (Archit et al., 2025) on a GPU-accelerated backend. This allowed histological experts to generate precise pixel-level masks by providing simple point-based prompts, dramatically reducing the annotation time compared to manual tracing.

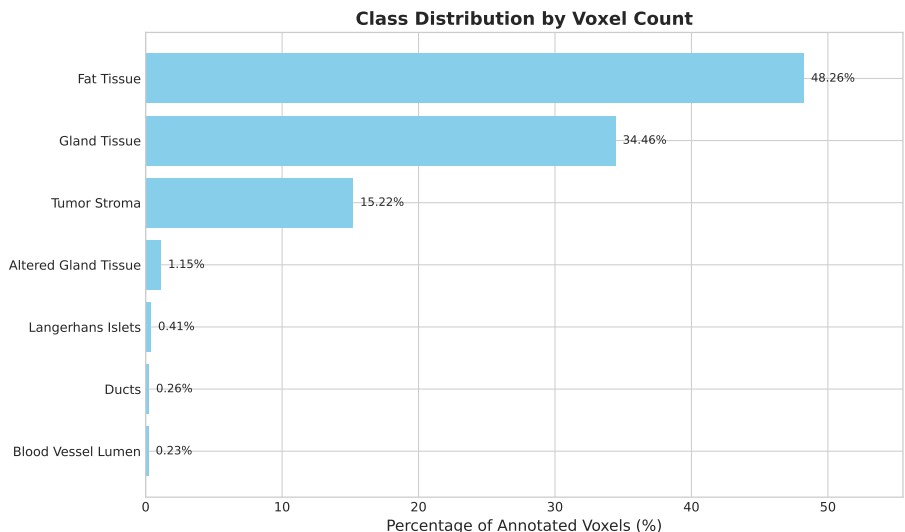

Figure 4: **Class Distribution by Voxel Count.** The dataset exhibits extreme class imbalance, with *Fat tissue* and *Gland tissue* accounting for over 82% of all annotated voxels. This motivates the use of a class-weighted loss function.

### A.2. Data Characteristics and Pre-processing

The resulting dataset is characterized by significant axial sparsity. Of the 2,460 total slices, only 772 (31.4%) contain any ground-truth annotations. A detailed analysis of these annotated regions, which sum to over 1.6 billion foreground voxels, reveals two further challenges for segmentation models.

First, the dataset exhibits a severe **class imbalance**, as illustrated in Figure 4. The two most dominant classes, *Fat tissue* and *Gland tissue*, account for over 82% of all annotated voxels, while critical structures like *Ducts* and *Blood vessel lumens* each constitute less than 0.3%. This imbalance necessitates a carefully weighted loss function during training.

Second, the presence of classes varies across volumes. A co-occurrence analysis (see Appendix A.3) reveals that several classes, including *Tumor Stroma* and *Gland Tissue*, appear in only one of the three volumes.

For network training, we employed an anisotropic downsampling strategy, reducing the in-plane XY resolution to 128x128 pixels while preserving the full depth dimension to retain all annotations. The intensity values of each volume were then normalized using z-score standardization.

**Class Co-occurrence Across Volumes**

| | Altered Gland Tissue | Fat Tissue | Blood Vessel Lumen | Tumor Stroma | Langerhans Islets | Ducts | Gland Tissue |
|---|---|---|---|---|---|---|---|
| Altered Gland Tissue | 1 | 0 | 1 | 1 | 1 | 1 | 0 |
| Fat Tissue | 0 | 2 | 1 | 0 | 1 | 0 | 1 |
| Blood Vessel Lumen | 1 | 1 | 2 | 1 | 1 | 1 | 0 |
| Tumor Stroma | 1 | 0 | 1 | 1 | 1 | 1 | 0 |
| Langerhans Islets | 1 | 1 | 1 | 1 | 2 | 1 | 1 |
| Ducts | 1 | 0 | 1 | 1 | 1 | 1 | 0 |
| Gland Tissue | 0 | 1 | 0 | 0 | 1 | 0 | 1 |

Figure 5: **Class Co-occurrence Heatmap.** The matrix shows the number of volumes (out of 3) in which each pair of classes appears together. The lack of universal co-occurrence makes a simple train/validation split of the volumes infeasible.

### A.3. Dataset Co-occurrence Analysis

To inform our training strategy, we performed a co-occurrence analysis to determine how frequently histopathological classes appear together within the same volume. The results are shown in the heatmap in Figure 5.

The matrix reveals that a simple train/validation split of the volumes is not viable. For example, *Gland Tissue* and *Tumor Stroma* do not co-occur in any volume. A split could therefore result in a validation set containing classes never seen during training, making a fair evaluation of generalization impossible. This finding justifies our methodological choice to pool all three volumes for the training process and to evaluate the model on a held-out fraction of annotated slices from across all volumes, as described in Section 4.4.

## Appendix B. Hallucination Analysis

To assess whether the continuity constraint induces false structure propagation, we computed a per-class hallucination rate: the fraction of annotated slices where a class is predicted but absent in the ground truth. Results are shown in Table 3.

Table 3: Per-class hallucination rate (percentage of annotated slices where a class is predicted but absent in the ground truth). The continuity loss dramatically reduces hallucinations across all major classes.

| Class | Baseline+Attn | Proposed | Change |
|---|---|---|---|
| Altered Gland Tissue | 0.53% | 0.00% | −0.53% |
| Fat Tissue | 0.00% | 0.26% | +0.26% |
| Blood Vessel Lumen | 4.76% | 4.50% | −0.26% |
| Tumor Stroma | 17.99% | 0.79% | −17.20% |
| Langerhans Islets | 3.97% | 1.32% | −2.65% |
| Ducts | 10.58% | 0.00% | −10.58% |
| Gland Tissue | 24.34% | 0.00% | −24.34% |
| **Overall** | **8.88%** | **0.98%** | **−7.90%** |

The continuity loss reduces the overall hallucination rate from 8.88% to 0.98%. The largest reductions occur for dominant classes (*Gland Tissue*: 24.34%→0%, *Tumor Stroma*: 17.99%→0.79%), where the baseline tends to over-predict boundaries into neighboring slices. Random, inconsistent predictions on individual slices are suppressed by the continuity constraint, as they are inherently incompatible with adjacent predictions. The sole minor increase (*Fat Tissue*: +0.26%) is negligible. These results confirm that the continuity loss acts as a stabilizing regularizer rather than a source of error propagation.

## Appendix C. Detailed Qualitative Results

To provide a more comprehensive qualitative assessment of our final model's performance, Figure 6 presents detailed comparisons of ground-truth annotations and the model's dense predictions on two representative volumes from our dataset.

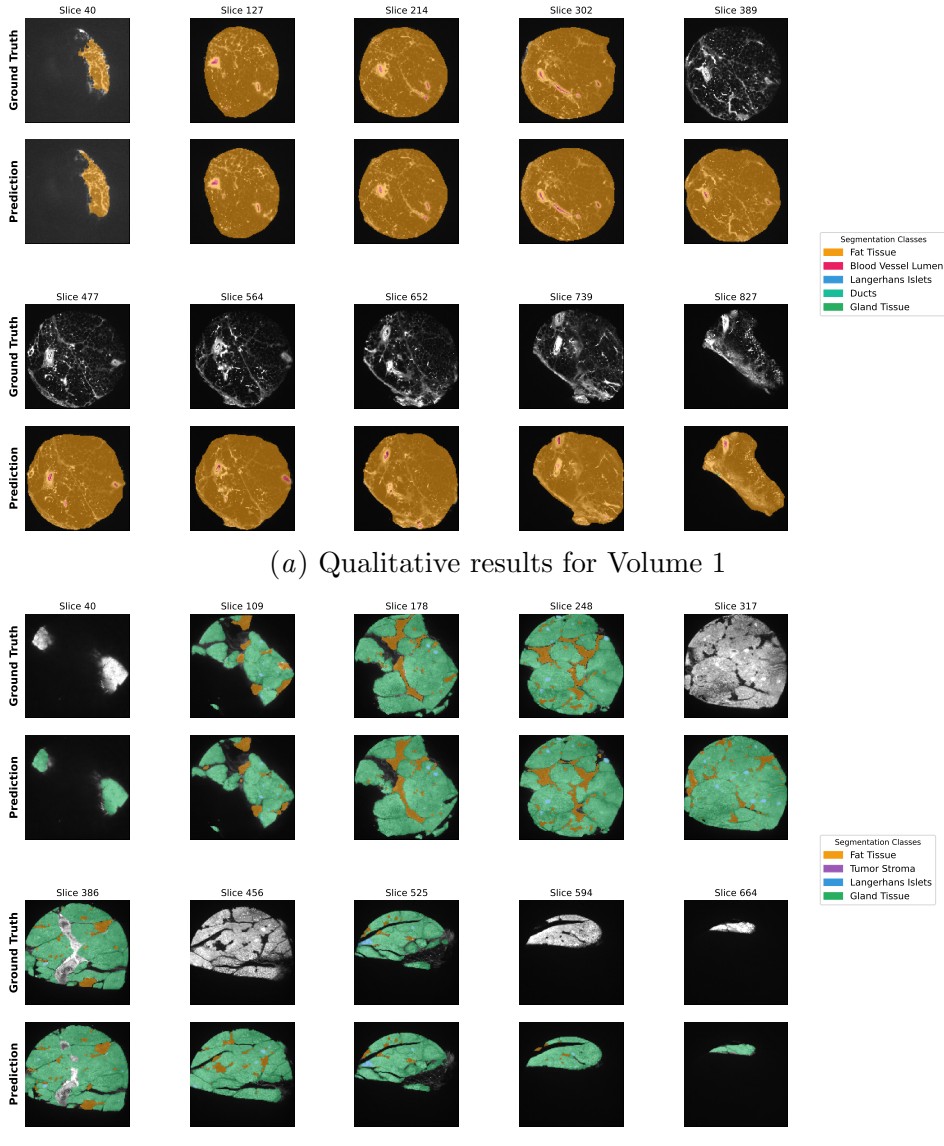

(*a*) Qualitative results for Volume 1

(*b*) Qualitative results for Volume 2

Figure 6: The model demonstrates high accuracy against the ground truth on annotated slices, while predictions on unannotated slices exhibit strong structural similarity and anatomical coherence, effectively bridging the gaps in supervision.

