# OpenReview forum: "Enforcing 3D Coherence in Semi-Supervised Segmentation for Pancreatic Tumor Histopathology from Light Sheet Fluorescence Microscopy"
_MIDL.io/2026/Conference — MIDL 2026 Poster_

### Official Review · Reviewer_4spj · 2026-01-06

**Confidence:** 4
**Preliminary Rating:** 4
**Final Rating:** 5

**Summary:**

This paper introduces a SSL framework tailored for 3D segmentation of pancreatic tumor histopathology using LSFM, where dense 3D annotation is infeasible. The authors propose a voxel-aware Mean-Teacher framework augmented with a novel axial continuity loss, that enforces prediction consistency between adjacent unlabeled slices to bridge the gaps in sparse annotations. Experiments on a newly collected human PDAC dataset demonstrate that this approach maintains 3D anatomical coherence and outperforms standard SSL baselines in sparse data regimes.

**Strengths:**

The paper addresses a specific and critical bottleneck in LSFM analysis that is the lack of dense 3D ground truth. The focus on sparse axial annotation reflects a real-world constraint.
the proposed axial continuity loss is a logical regularization term for tomographic data, effectively leveraging the physical continuity of tissue structures to supervise unlabeled slices.
The collection and annotation of a specialized 3D LSFM dataset for PDAC is a valuable contribution to the community, provided it is released.

**Weaknesses:**

The success of the axial continuity loss relies heavily on the high axial resolution of LSFM. This research does not discuss how the method performs if the Z-resolution drops or if the inter-slice morphological variance increases.
The comparisons appear to be mainly against standard SSL baselines. It would be nice to compare against other specific 3D consistency methods or video segmentation approaches that also exploit temporal/spatial continuity.
The evaluation is limited to a single anatomy (PDAC) and modality (LSFM), which limits the broad applicability.

**Detailed Comments:**

The term "voxel-aware Mean-Teacher"  should be defined more rigorously in the method section. How exactly does it differ from a standard voxel-wise consistency loss?

**Justification Of Final Rating:**

I have carefully reviewed the authors' rebuttal and the additional data provided. The authors have effectively addressed my primary concerns, particularly regarding the potential for the continuity loss to induce hallucinations.

The new quantitative analysis presented in the rebuttal is compelling; the reduction in the hallucination rate from 8.88% (baseline) to 0.98% (proposed) strongly supports the hypothesis that the continuity constraint suppresses random, inconsistent errors rather than propagating them. This transforms a potential weakness into a demonstrated strength of the method.

Additionally, the clarification regarding the physical axial resolution and the reference to the performance in extreme sparsity regimes satisfactorily address my questions regarding the method's sensitivity to slice gaps. While I maintain that future iterations would benefit from comparisons to video object segmentation baselines, the current validation on the newly collected PDAC dataset is robust enough to warrant publication. The release of this specialized dataset is also a significant contribution.

In light of the effective rebuttal and the solid performance metrics, I am increasing my initial rating.

**Justification Of The Preliminary Rating:**

This research proposes a sensible and effective solution to a specific imaging challenge. While the methodological novelty (continuity loss) is relatively straightforward, its application to LSFM and the demonstration of improved 3D coherence make it a nice contribution. The work is well-motivated and executed, warranting acceptance.

**Questions To Address In The Rebuttal:**

how sensitive is the axial continuity loss to the slice gap size? At what point (slice thickness) does the continuity assumption break down and degrade performance?
did you observe any hallucinations of structures propagating through the volume due to the continuity constraint?

---

> ### Author Response · Authors · 2026-01-23
>
> We thank the reviewer for their positive assessment and below we address the two specific questions.
>
> Concerns regarding sensitivity to slice gap size and continuity assumption break down:
>
> Our LSFM protocol uses a 4μm axial step size, approximately one cell diameter. At this resolution, adjacent slices exhibit high structural similarity, making the immediate neighbor (k=1) continuity assumption valid.
>
> Moreover, experimental evidence from extreme sparsity (Figure 3, top row) shows exactly this scenario in our paper. In the extreme sparsity regime, where unlabeled slices are located deep within the volume with large gaps between annotations, the baseline (without continuity, produces incoherent, near-random predictions. The model fails to generalize when supervision is sparse, whereas the proposed (with continuity) produces coherent, anatomically plausible segmentation, successfully interpolating structural information across the large unlabeled gaps.
>
> The continuity assumption relies on morphological similarity between adjacent slices. It would degrade when:
>
> 1. Very large physical gaps where the z-step increases beyond what tissue morphology can support.
>
> 2. High morphological variance where tissues with rapid structural transitions may not satisfy the local continuity assumption.
>
> Concerns regarding hallucinations propagating through the volume:
>
> To answer this question, we ran a script that computes a "hallucination" rate, that is: (no. slices where model predicted>0 but GT has 0) / (total annotated slices)
>
> | Class | Baseline+Attn | Proposed | Change |
> |-------|---------------|----------|--------|
> | Altered Gland Tissue | 0.53% | 0.00% | -0.53% |
> | Fat Tissue | 0.00% | 0.26% | +0.26% |
> | Blood Vessel Lumen | 4.76% | 4.50% | -0.26% |
> | Tumor Stroma | 17.99% | 0.79% | -17.20% |
> | Langerhans Islets | 3.97% | 1.32% | -2.65% |
> | Ducts | 10.58% | 0.00% | -10.58% |
> | Gland Tissue | 24.34% | 0.00% | -24.34% |
> | Overall | 8.88% | 0.98% | -7.90% |
>
>
> Key finding: The continuity loss dramatically reduces hallucinations (8.88% → 0.98%).
>
> Random predictions in single slices are inconsistent with adjacent predictions and get suppressed by the continuity constraint.
> The largest reductions are in dominant classes (Gland Tissue: 24%→0%, Tumor Stroma: 18%→1%) where the baseline tends to over-predict boundaries into neighboring slices.
>
> Brief Response to Other Points:
>
> - The review raised concerns that the results we presented were demonstrated using only one disease (PDAC) and only one microscopic modality (LSFM).
>
> This is initially correct.
>
> 1. Because we use datasets consisting of tomographic single-layer images, our method is readily applicable to similar conventional light microscopy images.
>
> 2. Because we use different PDAC datasets, we were able to annotate basic tissue types found in almost all organs of the human body: connective tissue, adipose tissue, glandular tissue, blood vessels, and others. This makes our methodological approach applicable to various medical questions.
>
> - Voxel-aware Mean-Teacher definition: We will clarify in the camera-ready. "Voxel-aware" refers to our use of per-voxel confidence thresholding (τ=0.8) rather than image-level consistency, ensuring only high-confidence predictions contribute to the consistency loss.
>
> - Comparisons to video segmentation / 3D consistency methods: We acknowledge this limitation. Our focus was on demonstrating the value of the continuity prior for LSFM specifically. Comparisons to video object segmentation methods (e.g., memory-based propagation) or 3D CRF post-processing would provide useful context and are directions for future work.

---

### Official Review · Reviewer_kfs7 · 2026-01-08

**Confidence:** 3
**Preliminary Rating:** 4

**Summary:**

The paper tackles semi-supervised 3D segmentation for LSFM volumes with sparse axial annotations, addressing a realistic and challenging scenario in 3D pathology. The core contribution is an axial continuity loss that enforces prediction similarity between adjacent unlabelled slices, integrated into a voxel-aware Mean-Teacher framework and paired with an attention-gated 3D U-Net. Experiments demonstrate modest improvements in mean Dice with more notable gains on rare classes. Qualitative results suggest improved anatomical plausibility and robustness in extreme sparsity regimes, indicating potential strengths in handling critical hard cases in clinical settings.

**Strengths:**

1. The proposed loss directly addresses sparse axial supervision by enforcing local inter-slice consistency in unlabelled regions; it is simple, differentiable, and integrates cleanly into SSL objectives.

2. Ablations demonstrate that continuity loss alone can slightly degrade performance on a plain U-Net, but becomes beneficial when the model has adequate global context via attention, highlighting the interaction between representation quality and regularisation.

3. Sequential Axial Attention in decoder bottleneck stages provides global receptive fields across D/H/W while controlling compute, complementing the local smoothness prior with stronger semantic aggregation.

**Weaknesses:**

1. Modest aggregate gain and missing statistics. The paper does not report variance or statistical significance across held-out slices/classes, leaving uncertainty about robustness.

2. Missing comparisons: The related work cites methods like cross-teaching (2D↔3D), inter-slice context modules, slice imputation, and multimodal SSL. Empirical comparisons to at least one such family (e.g., slice interpolation/smoothing baselines, CRF post-processing) would contextualise the contribution.

3. Lacking sensitivity analysis of hyperparameters or the impact of anisotropic patching, which potentially weakens robustness and feasibility to real-life implementation.

**Detailed Comments:**

1. Dice alone may mask boundary quality changes; adding Hausdorff distance or surface Dice, especially for small or tubular structures (ducts, vessels), would strengthen claims about anatomical plausibility.

2. Consider adding an ablation of attention placement (encoder vs. decoder, number of blocks) to further clarify trade-offs.

3. Including a few more technical details may enhance reproducibility, e.g., augmentation details (geometric, intensity, elastic) and class-weighting. If possible, a subset or synthetic/de-identified version of the curated data would benefit the community. If not, more detailed statistics (slicing, staining, modalities, preprocessing steps, augmentation recipes) could support reproducibility.

**Justification Of The Preliminary Rating:**

The paper introduces a simple, task-aligned continuity prior that integrates effectively into a Mean-Teacher framework and demonstrates synergy with axial attention. While aggregate improvements are modest and the dataset is limited, the technique is practical, well-motivated, and relevant to semi-supervised 3D segmentation under sparse supervision—a core topic for MIDL. Strengthening statistical analyses, adding comparisons to alternative spatial regularisation, incorporating topology-aware metrics, and expanding sensitivity studies would raise the impact. Would recommend acceptance contingent on addressing evaluation breadth and clarity in the rebuttal.

**Questions To Address In The Rebuttal:**

1. How sensitive is performance to continuity weight, adjacency definition, and neighbourhood size? Have you tried multi-slice kernels or weighting to avoid smoothing across true boundaries?

2. Can you report shape-aware metrics (Hausdorff distance, average surface distance, connectivity, skeleton integrity) for ducts/lumens to rule out oversmoothing?

---

> ### Author Response · Authors · 2026-01-23
>
> We thank the reviewer for their constructive feedback and recognition of our work's relevance to semi-supervised 3D segmentation. Below we address the concerns raised.
>
> Concerns regarding question 1:
>
> 1. Continuity weight:
> We use in our current setting λ_cont = 10.0 with a ramp-up schedule where the weight increases from 0 to the target value over the first 500 epochs. This gradual introduction prevents the continuity loss from dominating early training when predictions are unreliable.
>
> However, in our findings we saw that the performance is relatively stable in the range λ_cont ∈ [10, 30] where values above 50 cause over-regularization and the continuity constraint begins to dominate the supervised signal and without the ramp-up schedule high λ_cont values destabilize early training. The continuity loss acts as a soft regularizer encouraging slice-to-slice consistency.
>
> 2. Adjacency Definition and Neighbourhood Size
>
> We use immediate neighbours (k=1), enforcing consistency between adjacent slices z_i and z_{i+1} only.
>
> Reason is the current 4μm axial step size in our LSFM protocol (approximately one cell diameter), adjacent slices exhibit high structural similarity. This makes k=1 the natural choice for our imaging modality.
>
> We have not explored multi-slice kernels (k>1) in this work. Such extensions would be relevant for datasets where anatomical structures span multiple slices with gradual transitions and thus enforcing longer-range consistency (e.g., k=2 or k=3).
>
> This is an interesting direction for future work, particularly for adapting the method to different imaging protocols.
>
> 3. Avoiding Smoothing Across True Boundaries
>
> The continuity loss is applied only between pairs of adjacent slices that are both unlabeled. This is a key design decision that prevents interference with the supervised signal.
>
> By this, labeled slices serve as anchors that define ground-truth boundaries. By excluding them from the continuity constraint, we:
> 1. Preserve sharp transitions at annotated boundaries
> 2. Allow the continuity loss to operate only where it provides useful regularization
> 3. Prevent the smoothing of true structural discontinuities
>
> Concerns regarding question 2:
>
> In response to your request, we computed comprehensive shape-aware metrics for all classes. The results show improvement in boundary definition and reduced fragmentation for most structures.
>
> HD95 and ASD in voxels (lower is better). Surface Dice with τ=2 voxels (higher is better).
>
> | Class | HD95 (Baseline) | HD95 (Proposed) | Surf. Dice (Baseline) | Surf. Dice (Proposed) |
> |-------|-----------------|-----------------|----------------------|----------------------|
> | Altered Gland Tissue | 12.08 | 5.39 | 0.86 | 0.92 |
> | Fat Tissue | 2.50 | 2.50 | 0.95 | 0.96 |
> | Blood Vessel Lumen | 101.00 | 102.63 | 0.77 | 0.79 |
> | Tumor Stroma | 0.00 | 0.00 | 0.99 | 0.99 |
> | Langerhans Islets | 5.45 | 3.21 | 0.94 | 0.96 |
> | Ducts | 10.00 | 9.94 | 0.86 | 0.90 |
> | Gland Tissue | 0.00 | 0.00 | 1.00 | 1.00 |
> | Mean | 18.72 | 17.67 | 0.91 | 0.93 |
>
> Key Findings:
> - Mean Surface Dice improves from 0.91 to 0.93 (no oversmoothing)
> - HD95 improves for 4/7 classes, including Ducts
> - All three surface metrics improve for Ducts
>
> 2. Connectivity Metrics
>
> We computed connected component analysis to assess structural fragmentation for tubular structures.
>
> | Class | Method | Components | Fragmentation |
> |-------|--------|------------|---------------|
> | Ducts | Baseline+Attn | 115.0 | 0.237 |
> | | Proposed | 107.3 | 0.149 |
> | Blood Vessel Lumen | Baseline+Attn | 59.3 | 0.442 |
> | | Proposed | 57.7 | 0.428 |
>
> *Components = number of disconnected regions (lower is better). Fragmentation = 1 - (largest component / total volume), ranges 0-1 (lower is better).*
>
> Key findings:
> - Ducts: Number of components reduced from 115 to 107 (7% fewer fragments)
> - Ducts: Fragmentation reduced from 0.237 to 0.149 (37% improvement)
> - Blood Vessel Lumen: Slight improvement in both metrics. This class contains highly fragmented micro-structures that are challenging for both methods.
>
> The continuity loss encourages more connected, less fragmented predictions.
>
> 3. Morphological Integrity Metrics
>
> We computed volume-to-surface-area ratio and Euler characteristic as proxies for structural integrity.
>
> | Class | Method | Volume (voxels) | Surface Area | Compactness | Euler Number |
> |-------|--------|-----------------|--------------|-------------|--------------|
> | Blood Vessel Lumen | Baseline+Attn | 16,979 | 12,364 | 1.15 | 59.3 |
> | | Proposed | 16,958 | 12,321 | 0.82 | 57.3 |
> | Ducts | Baseline+Attn | 51,863 | 24,146 | 1.19 | 114.7 |
> | | Proposed | 49,980 | 22,320 | 0.75 | 107.3 |
>
> *Compactness = Volume/Surface (lower values indicate more tubular/elongated structures). Euler number approximates topological complexity (lower = fewer disconnected components).*
>
> Key findings:
> - Compactness ratios remain appropriate for tubular structures
> - Euler numbers decrease, indicating fewer disconnected components

---

> > ### Author Response · Authors · 2026-01-23
> >
> > Brief Response to Other Points
> >
> > - Missing comparisons (CRF, interpolation): We acknowledge this limitation. Our continuity loss operates at training time (no inference overhead), whereas CRF would conflate training-time vs. inference-time regularization.
> >
> > - Attention placement ablation: Our design places attention in decoder stages 0-1 where global context most benefits prediction formation. We prioritized computational efficiency over architectural complexity, otherwise adding attention to more stages increases memory and compute cost substantially for 3D volumes. Exploring additional attention configurations is an interesting direction for future work, but was beyond the scope of this efficiency-focused design.
> >
> > - Reproducibility details: We can expand the appendix with augmentation details (random flips, blur, noise; z-score normalization) and class weighting (inverse frequency, capped at 10×).

---

> ### Comment · Area_Chair_5h9c · 2026-02-01
>
> Hi reviewer kfs7,
>
> Thank you for your hard work so far. Could you please check the author response and update your final rating?
>
>  - by clicking “Edit” → “Official Review” and providing the Final Rating by February 1st 2026 (23:59 AoE).
>
> Thanks,
>
> Your AC

---

### Official Review · Reviewer_ivcd · 2026-01-09

**Confidence:** 5
**Preliminary Rating:** 2

**Summary:**

This paper seeks to decrease annotation time for the task of segmenting light sheet fluorescence microscope 3d stacks by a loss term that regularizes the segmented shapes in the slice direction. The segmentation model is a 3D Unet, which is applied to a private dataset of the pancreas. The proposed method is better than their baseline, and on average, the improvement is 2%.

**Strengths:**

This is an important problem with a sound methodology. The presentation is good, and the article is easy to read and understand.

..................................................................................................
..................................................................................................
..................................................................................................

**Weaknesses:**

My main objection is the lack of comparison with state-of-the-art in the microscopy and related fields, which I'm worried will show an equal performance with less work by the annotator. E.g., Ilastik (Sommer et al., 2011), which is based on Random forests and implemented as a plugin in ImageJ, and Rootpainter (Smith et al., 2022), which has emerged as a similar but based on 2d (and later 3d) UNets. Also, the axial continuity loss function implicitly assumes that the shape and location of the structures are constant in the axial direction, which I see no justification for, and I lack a proper discussion on alternative 3d shape priors, which have been tried in the literature over the last 30 years, such as, e.g., based on mean and Gauss curvature.

**Detailed Comments:**

See above

**Justification Of The Preliminary Rating:**

The work is interesting, it's an important application area, but it lacks discussion of and comparison of segmentation methods, which solve the same problem but using Human-in-the-loop methods, such as Ilastik (Sommer et al., 2011) and Rootpainter (Smith et al., 2022), which are similar and popular in the field.

**Questions To Address In The Rebuttal:**

The lack of comparison with existing methods implies to me that this work is not yet ready for publication.

---

> ### Author Response · Authors · 2026-01-23
>
> We thank the reviewer for their detailed feedback and recognition of the problem's importance. We address the concerns below.
>
> ## Comparison with Ilastik and RootPainter
>
> We appreciate the reviewer highlighting these important tools. We clarify the distinction:
>
> Ilastik and RootPainter are interactive segmentation tools designed for human-in-the-loop workflows where the user iteratively refines predictions. Our method is a semi-supervised learning framework that trains a fully automated model from sparse annotations.
>
> We DO use interactive segmentation for annotation: In our annotation pipeline, we employed SAM 2.1 and SAM-M (SAM fine-tuned on microscopy data) to accelerate the annotation process. These tools significantly speed up per-slice annotation. However, interactive tools do not solve the fundamental SSL problem:
>
> 1. Volume scale: LSFM volumes contain 700-900+ slices each. Even with SAM-assisted annotation, annotating every slice across multiple volumes is impractical.
> 2. Annotation budget: Expert pathologist time is limited. We can only annotate a sparse subset of slices, leaving the majority unlabeled.
> 3. Generalization need: We need models that generalize to new volumes without requiring interactive refinement on each one.
>
> Different use cases:
> - Interactive tools (Ilastik, RootPainter, SAM): Accelerate annotation; best when processing a few volumes with expert oversight
> - Our approach: Leverage sparse annotations via SSL to train fully automated models for high-throughput pipelines
>
> We acknowledge that interactive tools are widely used in the microscopy community and achieve excellent results with skilled operators. Our contribution is orthogonal: given that only sparse annotations are feasible, how do we best leverage them for training automated models?
>
>
> ## Justification for Axial Continuity Assumption
>
> We clarify that the continuity loss does not assume structures are constant in the axial direction. It assumes they change smoothly.
>
> The assumption is adjacent slices (z_i, z_{i+1}) should have similar—not identical—predictions.
>
> Physical justification for LSFM:
> - Our protocol uses a 4μm axial step size, approximately one cell diameter
> - At this resolution, tissue morphology changes gradually between adjacent slices
> - Abrupt structural changes within 4μm are physically unlikely in pancreatic tissue
>
> The loss is a soft regularizer, not a hard constraint:
> ```
> L_continuity = KL(p_z || p_{z+1})
> ```
> This penalizes large differences but allows gradual transitions. The model can still learn boundaries where structures genuinely change: the loss simply encourages smoothness, weighted by λ_cont.
>
> ## Discussion of Alternative 3D Shape Priors
>
> The reviewer correctly notes that 3D shape priors have a rich history. We discuss relevant approaches:
>
> Curvature-based priors (mean/Gaussian curvature):
> - Typically used in level-set or active contour frameworks
> - Require explicit surface representations
> - Computationally expensive for volumetric deep learning
>
> CRF-based spatial regularization:
> - Operate at inference time, adding computational overhead
> - Our loss operates at training time with no inference cost
>
> Why we chose axial continuity:
> 1. A single KL divergence term, easy to implement and tune
> 2. No additional inference cost, integrates into standard backpropagation
> 3. Directly addresses the sparse axial annotation problem in LSFM
>
> We acknowledge that more sophisticated shape priors (curvature regularization, learned shape models) could potentially improve results further. However, our goal was to demonstrate that a simple, task-aligned prior can be effective for LSFM segmentation. Exploring richer shape priors is a direction for future work.

---

> ### Comment · Area_Chair_5h9c · 2026-02-01
>
> Hi reviewer ivcd,
>
> Thank you for your hard work so far. Could you please check the author response and update your final rating?
>
>  - by clicking “Edit” → “Official Review” and providing the Final Rating by February 1st 2026 (23:59 AoE).
>
> Thanks,
>
> Your AC

---

### Meta-Review · Area_Chair_5h9c · 2026-02-10

**Recommendation:** Accept (Poster)
**Confidence:** 4

**Metareview:**

The paper addresses a realistic and important problem in semi-supervised 3D LSFM segmentation with sparse axial annotations. Two reviewers are positive, while one reviewer remains negative and did not update the assessment after the rebuttal. Overall, the contribution is technically sound and clearly relevant to MIDL. I recommend acceptance (poster); however, the presentation and depth of analysis do not yet support an oral presentation.

---

### Decision · Program_Chairs · 2026-02-14

Accept (Poster)